# Two-Dimensional Liquid Water Flow through Snow at the Plot Scale in Continental Snowpacks: Simulations and Field Data Comparisons

Ryan W. Webb[1,2,3], Keith Jennings[4,5,6], Stefan Finsterle[7], Steven R. Fassnacht[8,9,10]

[1]Department of Civil, Construction, & Environmental Engineering, University of New Mexico, Albuquerque, NM 87131 USA
[2]Center for Water and the Environment, University of New Mexico, Albuquerque, NM 87131 USA
[3]Institute of Arctic and Alpine Research, University of Colorado Boulder, Boulder, CO 80303 USA
[4]Lynker, Boulder, CO 80301 USA
[5]Department of Geography, University of Nevada, Reno, NV 89557 USA
[6]Desert Research Institute, Reno, NV 89512 USA
[7]Finsterle GeoConsulting, Kensington, CA 94708 USA
[8]Ecosystem Science and Sustainability - Watershed Science, Colorado State University, Fort Collins, CO 80523 USA
[9]Coopertive Institute for Research in the Atmosphere, Colorado State University, Fort Collins, CO 80521 USA
[10]Natural Resources Ecology Laboratory, Colorado State University, Fort Collins, CO 80523 USA

*Correspondence to*: Ryan W. Webb (rwebb@unm.edu)

**Abstract.** Modelling the multi-dimensional flow of liquid water through snow has been limited in spatial and temporal scales to date. Here we present simulations using the iTOUGH2 model informed by the model SNOWPACK, referred to as SnowTOUGH. We use SnowTOUGH to simulate snow metamorphism, melt/freeze processes, and liquid water movement in two-dimensional snowpacks at the plot scale (20 m) on a sloping ground surface during multi-day observation periods at three field sites in northern Colorado, USA. Model results compare well with sites below treeline and above treeline, but not at a site near treeline. Results show the importance of longitudinal intra-snowpack flow paths (i.e. parallel to ground surface in the downslope direction and sometimes referred to as lateral flow), particularly during times when the snow surface (i.e. snow-atmosphere interface) is not actively melting. At our above treeline site, simulations show that longitudinal flow can occur at rates orders of magnitude greater than vertically downward percolating water flow at a mean ratio of 75:1 as a result of hydraulic barriers that divert flow. Our near treeline site simulations resulted in longitudinal flow slightly less than vertically percolating water, and the below treeline site resulted in negligible longitudinal flow of liquid water. These results show the increasing influence of longitudinal intra-snowpack flow paths with elevation, similar to field observations. Results of this study suggest that intra-snowpack longitudinal flow may be an important process for consideration in hydrologic modelling for higher elevation headwater catchments.

## 1 Introduction

The presence, storage, and movement of liquid water within a snowpack has direct implications for land surface albedo (Dietz et al., 2012), wet-snow avalanches (Mitterer et al., 2011), streamflow generation (Hirashima et al., 2010; Wever et al., 2014), and rain-on-snow runoff generation (Würzer et al., 2016). During snowmelt and rain-on-snow events, the

movement of liquid water through snow is a major factor in controlling the timing and magnitude of runoff (Brauchli et al., 2017; Colbeck, 1972; Musselman et al., 2018; Würzer et al., 2016). Although liquid water flow is typically thought of as

acting primarily in the vertical direction, previous work has shown that intra-snowpack longitudinal flow can affect the timing, volume, and spatial patterning of runoff. We define intra-snowpack longitudinal flow as flow parallel to the ground surface in the downslope direction (sometimes referred to as lateral flow). Such intra-snowpack longitudinal flow has been shown to deposit runoff directly into streams, bypassing soil interaction (Eiriksson et al., 2013; Liu et al., 2004), and to create focused soil infiltration capable of altering runoff processes (i.e. produce infiltration excess; Webb et al., 2018d). Field observations

have shown intra-snowpack flow paths to range in scale from centimetres up to tens of meters (Avanzi et al., 2017; Kattelmann, 1985; Schneebeli, 1995; Webb et al., 2018a; Williams et al., 2010). However, modelling of this spatio-temporally complex process has been limited to one-dimensional (i.e. vertical) or centimetre-scale simulations (e.g. Wever et al., 2014; Würzer et al., 2017).

Multi-dimensional numerical models simulating preferential flow (vertical and/or longitudinal) through snow have

only been recently developed (Hirashima et al., 2019; Hirashima et al., 2017; Hirashima et al., 2014; Leroux and Pomeroy, 2017). These models apply long-understood soil physics using laboratory parameterization of snow properties (Calonne et al., 2012; Yamaguchi et al., 2010). As a result, these processes have been simulated primarily in centimetre-scale studies (e.g. Hirashima et al., 2019). However, there remains a need to understand these processes at the plot scale (multiple meters) to further understand the hydrological impacts, namely the amount of liquid water transported in the horizontal relative to the

vertical direction. Processes to consider during liquid water flow through snow include snow metamorphism, the melting of snow, and re-freezing of liquid water. These processes create temporally dynamic media properties, specifically snow grain size and porosity, creating a more complex environment relative to soil (Webb et al., 2018b). The layered characteristics of a snowpack and rapid metamorphism that occurs during melt (McGurk and Marsh, 1995; Marsh, 1987; Marsh and Woo, 1985) create temporary hydraulic barriers (Webb et al., 2018b) and thus temporally dynamic flow paths. Understanding and modeling

these dynamic flow paths at the plot scale remains an outstanding challenge in snow science.

The goal of this study is to advance the understanding of the spatio-temporal scales of longitudinal intra-snowpack flow paths by simulating liquid water flow through a layered snowpack at the plot scale. The research objectives are: 1) use the model SNOWPACK (Bartelt and Lehning, 2002) to simulate snow metamorphism, melting, and re-freezing processes; 2) utilize enhancements to the TOUGH2 (Pruess et al., 2012) non-isothermal multiphase flow and transport model, as

implemented in the iTOUGH2 simulation-optimization framework (Finsterle, 2020, 2017), to simulate water flow through a two-dimensional, temporally dynamic, layered snowpack at the plot scale; and 3) compare results to field observations under varying snowpack conditions.

## 2 Methods

We simulated liquid water flow through snow at three experimental plots, modelled as 20 m long two-dimensional domains with a hillslope angle of 10° (Fig. 1). Within this domain, the iTOUGH2 numerical model simulated the flow of liquid water with time-varying snow layer properties provided by the SNOWPACK model. It is important to note that the models were not fully coupled. For each iTOUGH2 time step, material properties were updated using output from SNOWPACK at that time step. For the remainder of this paper this soft coupling of SNOWPACK and iTOUGH2 will be referred to as SnowTOUGH. For SnowTOUGH testing, we limited the time domain of simulations to match field observations. We initiated the simulations during the first snow pit observations and ended them approximately three days later at the completion of experiments at each study plot.

### 2.1 SNOWPACK

The time-dependent material properties of the SnowTOUGH simulations were informed using the physically based SNOWPACK model (Bartelt and Lehning, 2002; Lundy et al., 2001). SNOWPACK discretizes the snow profile into layers, adding layers during accumulation events and consolidating them during compaction and melt. SNOWPACK closes the mass and energy balances at each time step and includes physically based routines for internal snowpack processes including energy exchange, snow grain metamorphism, and liquid water transport. SNOWPACK has been extensively validated in multiple environments and snow conditions (e.g. Jennings et al., 2018; Lundy et al., 2001; Meromy et al., 2015; Wever et al., 2016).

Simulations were run at hourly timesteps with quality-controlled meteorological observations. Air temperature, relative humidity, wind speed, incoming shortwave radiation, incoming longwave radiation, and snow depth data were used as forcing data for the SNOWPACK simulations. The SNOWPACK canopy module was activated for the below treeline study plot (described below) using physically representative values of leaf area index (4.0 $m^2$ $m^{-2}$), canopy height (7.0 m), and direct throughfall fraction (0.2, dimensionless). Liquid water transport was simulated using the default bucket scheme for full water year simulations and the Richards equation option (Wever et al., 2014) for simulating the intensive observation period (IOP) at each study plot. Full water year simulations were used to offer context to the timing of each IOP relative to peak snow water equivalent (SWE) and snowmelt processes (Fig. 2). For the IOP simulations, initial conditions were provided through manual snow pit observations (Webb et al., 2020; Webb et al., 2018c) so that we could focus our analysis on the intra-snowpack flow of liquid water and comparisons to field observations rather than the accuracy of the SNOWPACK simulated stratigraphy and the potential implications on our results. We revised input files for these simulations to be as consistent as possible with SNOWPACK's representation of measured and non-measured parameters. For example, we set liquid water fraction values based on snow pit observations (very wet/slush = 0.07, wet = 0.05, little water = 0.03, sticky = 0.01; Bradford *et al*., 2009; Techel & Pielmeier, 2011; Webb *et al*., 2018c). We also defined sphericity to be 1 and dendricity to 0 as most of the snow had metamorphosed by the time of observation, and we set grain types based on Lehning *et al.* (2002). We estimated bond radius as 30% of the measured grain radius. For more information on SNOWPACK simulations, see Webb et al. (2020; 2018c).

95          Snow layer variables were calculated by SNOWPACK at hourly intervals, specifically snow grain diameter ($d$), bulk snow density ($\rho_s$), volumetric liquid water content ($\theta_w$), and volumetric ice content ($\theta_i$). The dry density of each snow layer ($\rho_{ds}$) was calculated by multiplying $\theta_i$ by the density of ice (917 kg m$^{-3}$). The melt/freeze rate of each layer was determined by changes in $\theta_i$. Van Genuchten parameters (Van Genuchten, 1980) of unsaturated flow and water retention (i.e. $\alpha$ and $n$) were determined from SNOWPACK output using equations developed by Yamaguchi et al. (2012):

$$\alpha = 4.4(10^6)\left(\frac{\rho_{ds}}{d}\right)^{-0.98} \tag{1}$$

$$n = 1 + 2.7(10^{-3})\left(\frac{\rho_{ds}}{d}\right)^{0.61} \tag{2}$$

100         The intrinsic permeability ($K$) of each snow layer was defined using SNOWPACK output and the equation developed by Calonne et al. (2012):

$$K = 3.0r_{es}^2\exp(-0.013\theta_i\rho_i) \tag{3}$$

where $r_{es}$ is the equivalent sphere radius and $\rho_i$ is the density of ice.

## 2.2 iTOUGH2

            iTOUGH2 is a simulation-optimization framework for the TOUGH suite of numerical models that have been utilized
and validated for a range of processes in porous media (e.g. Fujimaki et al., 2008; Hannon and Finsterle, 2018; Ho and Webb, 1998; Kechavarzi et al., 2008; Lippmann and Bodavarsson, 1983). For this study we used the equation of state module 9 (EOS9), applying Richards' equation (Richards, 1931) for the transport of liquid water only and does not consider energy transport (Pruess et al., 2012). We used new enhancements to the iTOUGH2 code (Finsterle, 2018) that allow for time-dependent material properties and time-dependent material-related source/sink terms to simulate the snow metamorphism and
melt/freeze processes in a layered snowpack. The layered snowpack was modelled above a 10–30 cm deep soil, increasing in depth under deeper snow. Deeper soil was modelled under deeper snow to increase pore storage volume available for any infiltrating water released from the snowpack. The boundary conditions of the upslope and downslope ends of the domain were simulated as no-flow conditions and a drain was modelled at the downslope end to remove excess liquid water that may build up on the no-flow boundary (Fig. 1). Soil types for each site are known as silty loam and retention parameters common
to this soil type were used. These parameters were a van Genuchten $m$ value of 0.29, a porosity of 0.67, and van Genuchten $\alpha$ value of 0.02 cm$^{-1}$. The slightly high porosity was chosen to allow for additional soil moisture storage, if necessary. Saturated hydraulic conductivity estimates of the soils were taken as the mean of more than 15 mini-disc infiltrometer observations distributed evenly across a 10 m × 20 m plot at each site. These saturated hydraulic conductivity values for the below treeline (BT), near treeline (NT), and above treeline (AT) sites were $1.36 \times 10^{-3}$ cm s$^{-1}$, $6.93 \times 10^{-4}$ cm s$^{-1}$, and $8.46 \times 10^{-4}$ cm s$^{-1}$,
respectively. The model was discretized into elements 25 cm in length and 1 cm in height (Fig. 1). Similar to SNOWPACK simulations used for the IOP, initial conditions were provided through manual snow pit observations (Webb et al., 2020; Webb et al., 2018c).

The material properties that we defined to vary through time were permeability, the van Genuchten $m$ term ($m = 1 - 1/n$), and van Genuchten $\alpha$ term. The time-dependent source/sink terms were used to simulate the melt/freeze processes. The

melt rate of snow and re-freezing of liquid water, determined from SNOWPACK, was used to quantify time-dependent source terms for liquid water introduction via snowmelt and as corresponding sink terms for the re-freezing of water. The movement of liquid water simulated by SnowTOUGH was then compared to field observations at the three study plots.

Flowrates for each simulation were calculated for a 1 m × 1 m footprint of hillslope. Thus, longitudinal flow is for a 1 m wide section of hillslope summed over the entire depth of the snowpack and vertical flow is for a 1 m$^2$ area on the ground

surface summed over the entire depth of the snowpack. These calculations were conducted at a location 15 m downslope and as an average per snow profile for 10 m upslope from this location. Bulk $\theta_w$ values were also calculated for all snow profiles for this same area between 5 m and 15 m downslope for SnowTOUGH simulations to compare to field observations. SnowTOUGH results were analysed for this 10 m length of hillslope to eliminate boundary effects of the upslope and downslope boundary conditions on analyses.

**3 Field Sites**

Field observations of snowpack $\theta_w$, stratigraphy, and longitudinal flow paths were conducted at three locations in the Colorado Rocky Mountains using a combination of dye tracer experiments and ground-based remote sensing. These sites ranged in elevation from a north facing BT site at 2700 masl in a lodgepole pine forest, a south facing NT site at 3350 masl in a large forest clearing, and a southeast facing AT site at 3500 masl. Each site had continuous observations of snow depth and

air temperature with additional meteorological stations at the AT and BT sites. All sites had a ground surface slope of ~10°. For more information on site descriptions and meteorological data, see Webb et al. (2020; 2018c).

During the IOPs, which we timed to occur near peak SWE, a dye tracer (Rhodamine WT) was applied at each of the study plots immediately prior to the first snow pit observation. This was the time of initiation for SnowTOUGH simulations. The dye tracer was subsequently allowed to move into and through the snow undisturbed for at least two full days prior to a

second set of snow pits being dug downslope of application. These snow pits allowed us to observe locations of longitudinal flow paths that transported the dye tracer. For further information concerning the dye tracer experiments see Webb et al. (2020).

Additionally, ground-based remote sensing techniques were used to estimate the plot-scale distribution of bulk $\theta_w$ multiple times throughout each IOP for the NT and AT sites. Snow depths derived from terrestrial LiDAR scanning were collected in combination with ground penetrating radar surveys to obtain spatially distributed dielectric properties of the

snowpack that were used to estimate the spatial distribution of bulk snowpack $\theta_w$. For further details of these ground-based remote sensing methods see Webb et al. (2018c).

## 4 Results

### 4.1 SNOWPACK

SNOWPACK output shows the progression of liquid water development, snow layer metamorphism, and melt/freeze processes at our three sites on both the seasonal timescale (Fig. 2) and during the IOP (Fig. 3). The timing of our observations was such that we captured a period early in the melt season for each plot. Spring snowmelt, defined to begin on the first simulated day of persistent liquid water in the snowpack in the full season simulations, began on March 3, March 18, and May 4 for the BT, NT, and AT study plots, respectively (Fig. 2). The first day of the IOP for each plot was March 5, April 12, and May 15 for the BT, NT, and AT study plots, respectively. The BT plot was the only site that was not actively melting and releasing liquid water immediately prior to our IOP while the NT site was melting for the longest time prior to our observations (Fig. 2a). The IOP at the AT site captured the onset of a storm and re-freezing of the snow surface (Fig. 3c). Thus, these three sites and IOPs captured varying stratigraphy and conditions that occur within any given melt season in mountainous environments.

During the IOP simulations, SNOWPACK results show little metamorphism with majority of the grain size changes occurring near the snow-atmosphere interface as a result of surface melt (Fig. 3). All three site simulations expressed diurnal melt cycles, including the retention of liquid water overnight. The BT simulations resulted in diurnal melt cycles that ranged from 0-1.1 mm hr$^{-1}$. The NT simulations resulted in melt rates ranging from 0-3.9 mm hr$^{-1}$, and the AT simulations resulted in melt rates ranging from 0-2.7 mm hr$^{-1}$. Additionally, the NT and AT SNOWPACK simulations display the occurrence of hydraulic barriers that persist throughout the entire melt season (Fig. 3). These results also highlight the increased stratigraphy and formation of ice lenses that occur at higher elevations and impact liquid water flow processes (e.g. Webb et al., 2018b). Thus, the IOP SNOWPACK simulations resulted three different snowpack conditions that varied in melt-freeze cycles, metamorphism, and snow accumulation/disappearance throughout the profiles.

### 4.2 SnowTOUGH

SnowTOUGH incorporated the melt/freeze processes and temporally dynamic snow layer variables into a two-dimensional plot-scale model. At the AT and NT sites, SnowTOUGH simulated the presence of multiple hydraulic barriers, holding vertically percolating water and transporting it longitudinally downslope (Fig. 4). Conversely, the BT simulations simulated minimal longitudinal liquid water flow (mean <0.1 ml s$^{-1}$), though increased water retention of vertical flow in specific layers did occur as a result of the layer water retention properties. During the IOP, mean vertical flux of water at the BT site was 1.5 ml s$^{-1}$ (Fig. 5). This dominance of vertical flow contrasts with the simulations of the two higher elevation snowpacks, which display the higher occurrence of hydraulic barriers as a result of the more complex stratigraphy (Figs. 3 & 4). Furthermore, soil infiltration was minimal at all three sites during the IOPs (Fig. 5) indicating that the majority of vertically moving liquid water was either held within the pore space of a snow layer or transported longitudinally. For all study sites, the

vertical flow of liquid water displayed negligible differences between locations at 5 m and 15 m downslope. Conversely, longitudinally diverted flow accumulated along flow paths for the entire length of the hillslope (Fig. 5).

For the NT site, peak longitudinal flowrates occurred just prior to 12:00 each day (Fig. 5b). The vertical flow of water was more variable than longitudinal flow at the NT site, with a mean of 9.1 ml s$^{-1}$ and a standard deviation of 8.5 ml s$^{-1}$. The longitudinal flow remained relatively steady at the 15 m downslope profile with a mean rate of 6.6 ml s$^{-1}$ and a standard deviation of 1.4 ml s$^{-1}$. Infiltration into the soil at the NT site showed a mean value of 0.7 ml s$^{-1}$ with a standard deviation of 0.5 ml s$^{-1}$. Longitudinal flow at the NT site occurred within a single longitudinal intra-snowpack flow path near the snow-soil interface.

The AT site had unique meteorological conditions during the IOP relative to the other sites. An incoming storm resulted in all surface melt halting during the afternoon of May 15 (Fig. 6). The snow surface temperature later warmed back to 0°C for a two-hour period with minimal surface melt simulated to occur during this brief time (1.2 mm). Though surface melt stopped, liquid water continued to flow for the entirety of the simulation, predominantly in the longitudinal direction (Figs. 5c & 6). The vertical flow of water at the AT site decreased rapidly as the storm moved in. For the entire IOP simulation, the AT site resulted in mean vertical flowrate of 5.8 ml s$^{-1}$ with a standard deviation of 6.6 ml s$^{-1}$. At the 15 m downslope profile, the AT simulations resulted in mean longitudinal flowrate of 212 ml s$^{-1}$ with a standard deviation of 47 ml s$^{-1}$. Soil infiltration for the AT site showed a mean value of 0.5 ml s$^{-1}$ with a standard deviation of 0.3 ml s$^{-1}$. SnowTOUGH simulations of the AT site resulted in three longitudinal intra-snowpack flow paths. The simulated number of intra-snowpack longitudinal flow paths for the AT and BT sites were equal to field observations whereas the NT site simulations did not match well with field observed locations of the dye tracer during experiments described in Webb et al. (2020).

### 4.3 Comparison to Field Data

Comparisons of SnowTOUGH to field observations indicated varying results based on site and parameter of interest. The simulated bulk $\theta_w$ showed little temporal variability for both the NT and AT sites while field observations showed greater variability (Fig. 7). Simulated bulk $\theta_w$ remained near 3% for both the NT and AT sites throughout the IOPs. The average of all field observations was generally greater than simulated values, though simulated values were always within one standard deviation of field observations. The mean of all field observations for the NT and AT sites was 4.2% and 3.5%, respectively. The large standard deviations of field observations were largely driven by converging intra-snowpack flow paths creating areas of bulk $\theta_w$ as high as 20% (Webb et al., 2020). Though different, the comparison of $\theta_w$ between SnowTOUGH simulations and field observations are within the estimated error of the field methods (~2%).

Dye tracer experiments compared well for the AT and BT sites, but not the NT site (Fig. 4). The number of intra-snowpack flow paths shown in the SnowTOUGH simulations were similar to those shown from dye tracer experiments at the AT (3 longitudinal flow paths) and BT (0 longitudinal flow paths) sites presented in Webb et al. (2020) though the depths of these flow paths beneath the snow surface differed slightly. The field-observed dye tracer locations that were not simulated as longitudinal flow paths for some instances still display increased liquid water retention or ice lens formation (Fig. 6).

Additionally, simulation results suggest increasing longitudinal fluxes of liquid water with elevation that was similarly observed at these sites (Webb et al., 2020). The largest discrepancy between simulations and field observations occurred at the NT site where only one intra-snowpack flow path was simulated using SnowTOUGH and three were observed in the field at different depths beneath the snow surface (Fig. 4).

Comparing simulated longitudinal flowrates to field observations is difficult because no flowrates were directly measured in the field. However, locations of converging intra-snowpack flow paths were used in Webb et al. (2020) to estimate effective upslope contributing areas (EUCAs), defined as the minimum upslope contributing area required to produce observed changes in liquid water content from melt rate estimates if all meltwater was diverted longitudinally and collected in a single observation location. Therefore, these observations are not directly comparable to SnowTOUGH simulations, but insights can still be gained from comparisons. These field observations resulted in peak EUCA of 6 $m^2$ and 17 $m^2$ for the NT and AT sites, respectively. For the NT site, this occurred over a two-hour time period with a total of ~5 mm of melt. At this same time in the NT SnowTOUGH simulations, the location 15 m downslope resulted in an accumulation of longitudinal flow of 28.7 mm of water (Fig. 5bii: 12-Apr 12:00, mean longitudinal flow of ~4 ml $s^{-1}$ for two hours), or roughly 5.7 times greater than the simulated melt. For the AT site, the simulated melt was ~7.5 mm with a total accumulation of simulated longitudinal flow, at the 15 m downslope location, of 386 mm (Fig. 5cii: 16-May 12:00, mean of 268 ml $s^{-1}$ for two hours), or roughly 51 times greater than the estimated contributing melt. Therefore, relative to the calculations from Webb et al. (2020), SnowTOUGH simulated longitudinal fluxes are similar to observations at the NT site and greater by a factor of three at the AT site.

## 5 Discussion

This is the first study, to our knowledge, that has simulated the two-dimensional flow of liquid water through a snowpack at the experimental plot scale. Our simulations show the presence of hydraulic barriers that divert liquid water longitudinally via preferential flow paths at the two upper elevation sites (AT and NT) that were also present in field observations.

SnowTOUGH simulations produced the greatest rates of longitudinal flow at the highest elevation site, the AT plot, similar to field observations using dye tracer experiments and ground-based remote sensing techniques (Webb et al., 2020). The number of longitudinal flow paths observed in the field at this site were equal to simulations. The depths of these flow paths beneath the snow surface however, differed slightly between field observations and simulations. This is likely a result of SnowTOUGH simulations not accounting for snow depth variability across the plot and natural snowpack layer heterogeneity (Leroux and Pomeroy, 2017; Marsh and Woo, 1985; Molotch et al., 2016). At the NT plot, some of the field-observed dyed flow paths were observed as dyed ice lenses (Webb et al., 2020) where SnowTOUGH simulated liquid water storage increased and ice lenses formed with no longitudinal flow. This may result in a longitudinal flow path during later melt events, though further experiments with longer IOPs are necessary. The natural heterogeneity of snowpack stratigraphy would be difficult to characterize at this scale without disturbing the snow at the location of the dye tracer experiment. Maintaining undisturbed

conditions is essential to study natural transport of the dye. Additional studies are necessary to characterize the horizontal heterogeneity of stratigraphy in varying snowpack conditions. Previous snow studies have suggested that the discontinuity of layers (such as ice lenses) can be a major factor in flow path continuity (Eiriksson et al., 2013; Kattelmann and Dozier, 1999; McGurk and Marsh, 1995; Schneebeli, 1995; Yamaguchi et al., 2018). However, previous studies of capillary barriers at the interface between soil layers have shown that homogenous layer assumptions, as those made in the present SnowTOUGH simulations, capture the average of randomized heterogeneous simulations (Ho and Webb, 1998). The validity of this assumption for snow should be further studied. In general, it is likely that the natural heterogeneity of both permeability and capillary barriers will decrease the amount of longitudinal flow simulated in this study. Thus, SnowTOUGH simulations are likely overestimating the amount of longitudinal flow for specific flow paths.

Relative to estimates of EUCA, the AT site simulations overestimated longitudinal flow. However, it is important to note that the field methods used to estimate the EUCA likely underestimate the value because it assumes all diverted liquid water remains in the snowpack at the point of calculation. Additionally, the low melt rates as a result of the incoming storm add uncertainty to the appropriateness of these calculations using snowmelt rates. Therefore, it is likely that the true value of EUCA is between 17 m$^2$ and 51 m$^2$ for the AT site. Conversely, the NT site simulations resulted in a similar amount of longitudinal flow within the plot-scale simulations as field observed EUCA suggests. Considering the underestimation of the number of flow paths simulated at this site and the underestimation of EUCA from field methods as previously mentioned, the true EUCA is likely larger than the SnowTOUGH-simulated longitudinal flow. Additionally, the longitudinal flux for the single flow path is likely overestimated. We recommend the use of snow lysimeters similar to those implemented in Eiriksson et al. (2013) in future studies to further quantify intra-snowpack longitudinal flow for comparison to the SnowTOUGH model.

The continued flow of liquid water after surface melt ceased at the AT site provides insights towards the movement of liquid water during the melt season. The flow paths continued to direct liquid water longitudinally downslope 40 hours after surface melt ceased (Fig. 4) and was confirmed during dye tracer collection and ground-based remote sensing observations. The dyed flow paths still contained liquid water at the time of field observations, implying that further longitudinal flow would have likely occurred for an uncertain time and distance. Furthermore, multiple layers retained liquid water that will be more readily available for transport during later melt events (Fig. 6). During this time of no surface melt, little vertical movement of water towards the ground surface occurred in the SnowTOUGH simulations and the dominant flow direction was longitudinal as a result of hydraulic barriers. The simulated longitudinal movement of water through the snowpack was orders of magnitude greater than vertical downward movement of meltwater with mean values of 212 ml s$^{-1}$ and 5.8 ml s$^{-1}$, respectively (mean ratio of 75:1, Fig. 5cii). Though this ratio is likely overestimated as previously mentioned, we estimate that the order of magnitude is correct. This suggests that during regular diurnal melt cycles in the spring snowmelt season, meltwater may continue flowing downslope overnight or during cold periods, accumulating at downslope convergent locations. For the AT plot, this location is where the ground surface and snow surface slope gradients decrease as was observed to accumulate liquid water in the ground-based remote sensing observations (Webb et al., 2020, 2018c) and produced large variability in snowmelt lysimeter discharge in previous years (Rikkers et al., 1996). This process also has implications for SWE distribution during mid-winter

melt events. Mid-winter melt events may initiate flow paths that divert liquid water along longitudinal flow paths with no infiltration across the snow-soil interface. This meltwater may flow downslope for many hours due to the relatively slow re-freezing process, accumulating flow at convergent locations prior to re-freezing. As a result, the distribution of SWE may be such that increased bulk density occurs at downslope locations of convergent flow paths with no obvious increase in depth, and potentially a decrease in depth. For example, Webb et al. (2018a) observed a 170% increase in SWE with a decrease in snow depth as a result of increased liquid water content from upslope locations.

The SnowTOUGH simulations bring new modelling capabilities of two-dimensional liquid water flow through snow. To date, multi-dimensional modelling has been limited to the centimetre scale (e.g. Hirashima et al., 2017; Leroux and Pomeroy, 2017). For simulations at the scale of meters to tens of meters, as presented in the current study, variable parameterization remains a challenge. Current hydraulic variables for snow layers have been developed at the centimetre scale in controlled laboratory environments (Calonne et al., 2012; Yamaguchi et al., 2010) and shown improvement for one-dimensional (vertical) models on flat terrain (e.g. Wever et al., 2016;  Wever et al., 2014). However, difficulties arise with layer heterogeneity as previously mentioned, sloping terrain as presented in this study, and if snow grain types vary (Yamaguchi et al., 2012). While these variable parameterizations worked well for the BT and AT study plots to identify longitudinal flow paths, the NT site simulations did not match field observations well, indicating the variables do not work as well when hydraulic barriers of lesser strength are present (i.e. smaller differences in layer properties across interfaces). Future studies at the plot scale may improve effective parameterization of specific layer variables through the application of snowmelt lysimeters and inverse modelling techniques. Improved parameterization of snow variables for modelling liquid water flow through snow would likely improve the modelling accuracy for hydraulic barriers that dominate liquid water transport during times of little or no surface melt (Fig. 4). These hydraulic barriers cause longitudinal flow paths at the plot scale and control the storage and release of liquid water. A logical next step for future studies aiming to model this process is to develop a fully coupled two-dimensional model and build upon parameterization of variables to determine effective properties at the plot to hillslope scales and the implications for hydrological modelling.

Current hydrological models do not account for longitudinal intra-snowpack flow paths. Snowmelt is assumed to vertically percolate through the snow and infiltrate the soil at the same location it originated as melt. However, the present study indicates that longitudinal intra-snowpack flow (i.e. lateral flow in hydrologic modelling terms) can be a dominant process at the plot scale. Given the higher hydraulic conductivities of snow relative to common soils (Calonne et al., 2012), the flow paths identified in our study may have important implications towards headwater catchment dynamics during the snowmelt period. These processes may be of particular interest if models are to be used in a predictive manner for future meteorological scenarios (i.e. climate warming).

## 6 Conclusions

Through the soft coupling of SNOWPACK and iTOUGH2, we successfully simulated the two-dimensional movement of liquid water through a layered snowpack, including snow metamorphism and melt/freeze processes, at spatial scales previously unstudied. The simulations compared well with field data at two of the three field sites. Results show the importance of longitudinal intra-snowpack flow paths, particularly during times when the snow surface re-freezes. We show the importance of longitudinal flow paths at the multiple meter scale and for temporal scales beyond regular diurnal fluctuations. At the above treeline study site, the longitudinal flow was orders of magnitude greater than vertically downward percolating water with a mean ratio of 75:1. At the near treeline site, longitudinal flow was simulated as slightly less than vertically downward percolating water. The below treeline simulations resulted in negligible longitudinal flow. This study shows the increasing influence of longitudinal intra-snowpack flow paths at higher elevations, where a snowpack develops a more complex and persistent stratigraphy. Results of this study suggest that intra-snowpack longitudinal flow may be an important process for consideration of streamflow timing in snowmelt dominated hydrographs for hydrologic modelling purposes.

## Acknowledgements

We would like to acknowledge the contributions of Guillaume Chambon, Nander Wever, and Hiroyuki Hirashima for their constructive feedback that was offered towards an earlier version of this manuscript. Additionally, we would like to acknowledge the contributions of the staff at the Niwot Ridge Long Term Ecological Research site and the Boulder Creek Critical Zone Observatory. This work was supported by the National Science Foundation award 1624853, the Niwot Ridge LTER cooperative agreement number DEB-1637686, and Boulder Creek CZO cooperative agreement number EAR-1331828.

## Data Availability

Availability of field collected data are as follows: LiDAR data are available through UNAVCO https://tls.unavco.org/projects/U-060/; GPR data are available at datadryad.org. SNOWPACK configuration, input, and output files can be found at http://dx.doi.org/10.17632/9wjjvsy82p.1.

## Author Contributions

RWW conceived of the study, collected field data, conducted SnowTOUGH model simulations, and analysed results. KJ conducted SNOWPACK simulations. SF modified TOUGH2 code and advised RWW during model simulations. SRF mentored RWW during study conceptualization. RWW led the writing and editing of the manuscript with contributions from all co-authors.

**Competing Interests**

The authors declare that they have no conflict of interest.

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

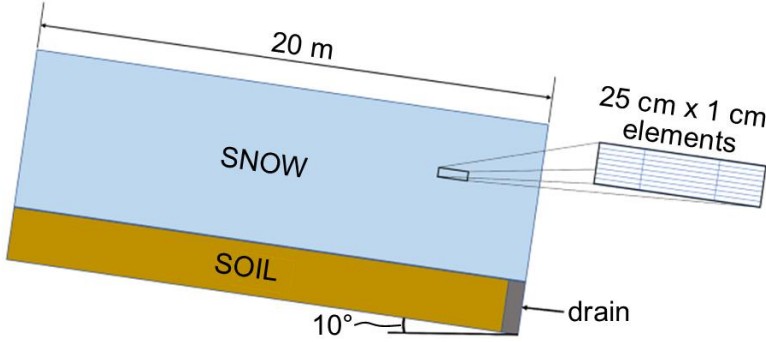

**Figure 1: Conceptual diagram of the model domain showing the snowpack above soil, the location of the drain in the soil, and element discretization. Figure not to scale.**

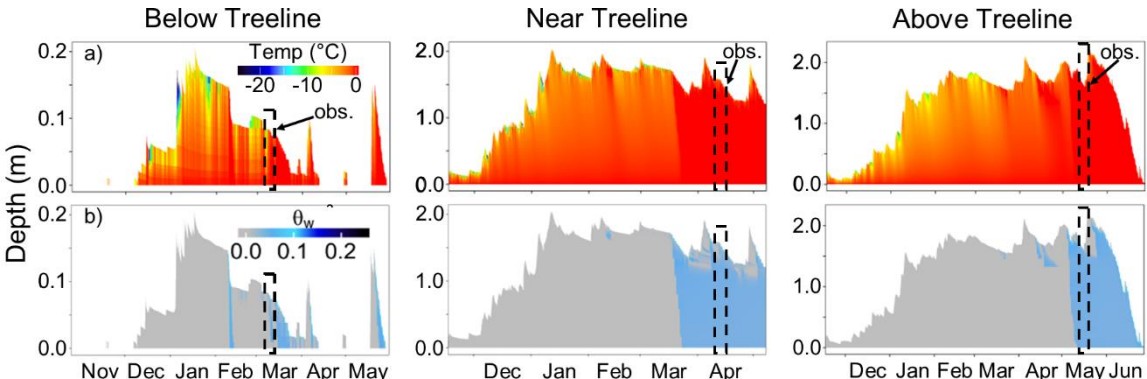

**Figure 2: Results of SNOWPACK simulations of the entire water year for below treeline (BT), near treeline (NT), and above treeline (AT) study sites. Results shown are a) snow temperature and b) volumetric liquid water content ($\theta_w$) with the observation (obs.) and SnowTOUGH simulation period highlighted in the dashed box. Note that the axes have different scales for each site and that these are not used as input into the SnowTOUGH simulations but used for context towards the timing of observations to seasonal processes.**



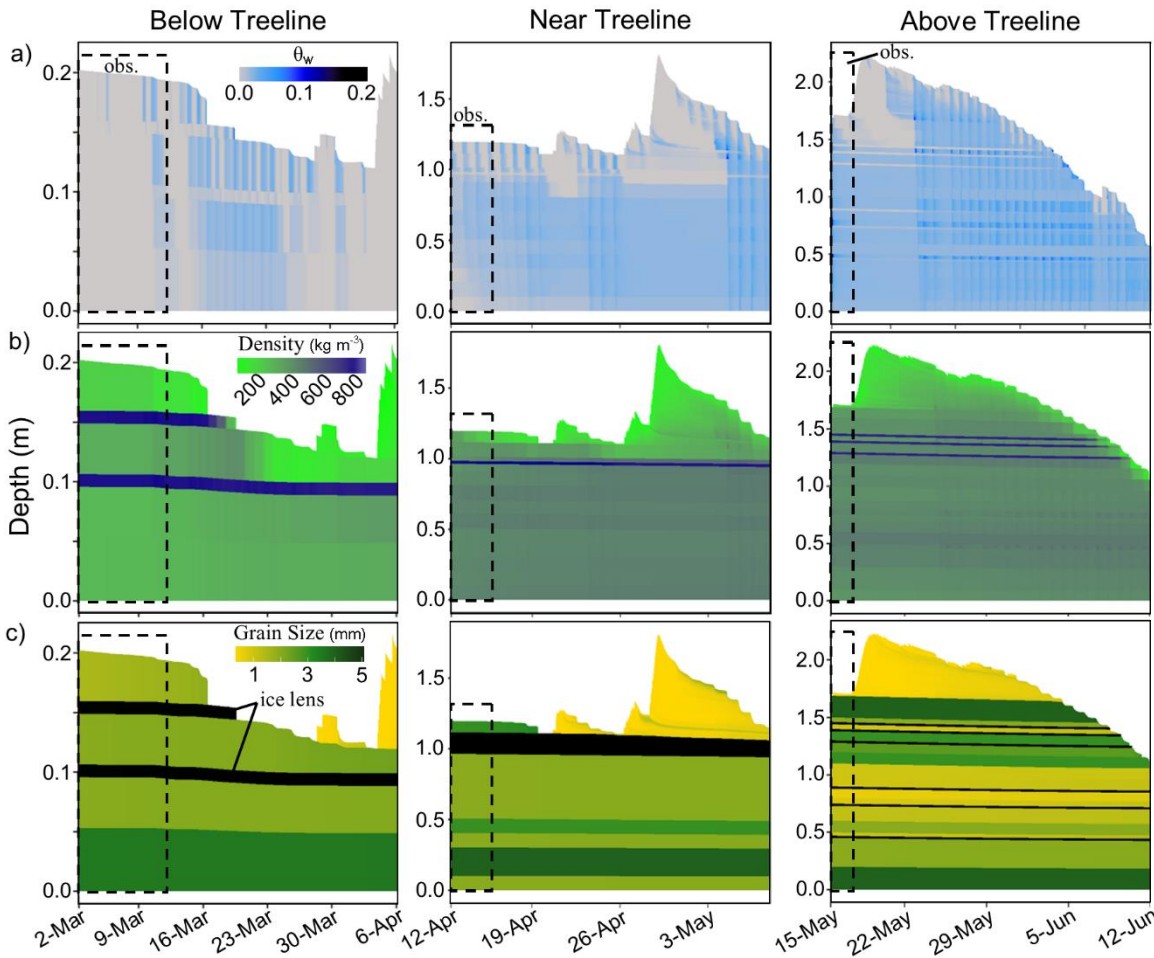

**Figure 3: Results of SNOWPACK simulations of the intensive observation periods (IOPs) for below treeline (BT), near treeline (NT), and above treeline (AT) study sites. Results shown are a) volumetric liquid water content ($\theta_w$), b) snow density, and c) grain size with the observation (obs.) and SnowTOUGH simulation period highlighted in the dashed box. Note that the axes have different scales for each site.**

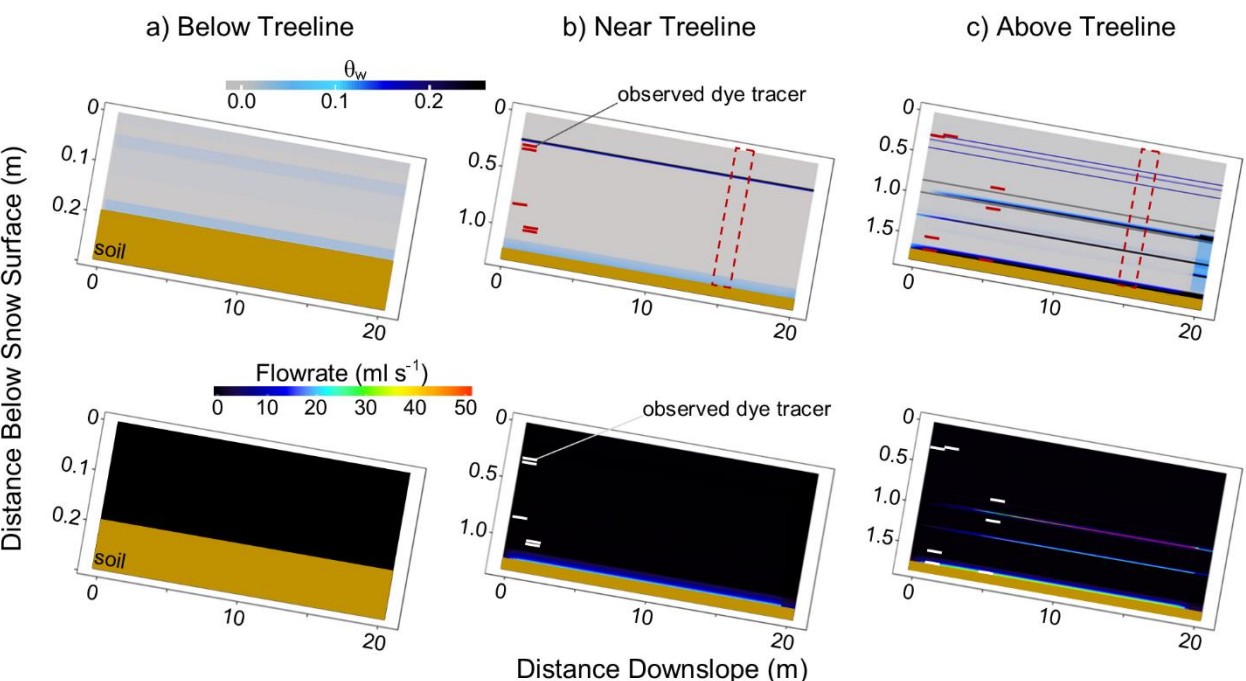

**Figure 4: Results of SnowTOUGH simulations for the study sites: a) below treeline (BT) at time = 3-Mar 06:00, b) near treeline (NT) at time = 14-Apr 04:00, and c) above treeline (AT) at time = 15-May 23:00. Results shown are (top) volumetric liquid water content ($\theta_w$) distribution and (bottom) liquid water flowrates in the longitudinal (i.e. parallel to ground surface) direction. The soil elements within the model domain are shown as solid brown to focus results on intra-snowpack distributions. Dashed red boxes in the $\theta_w$ plots indicate the profile of model elements used for flowrate calculations at 15 m downslope.**


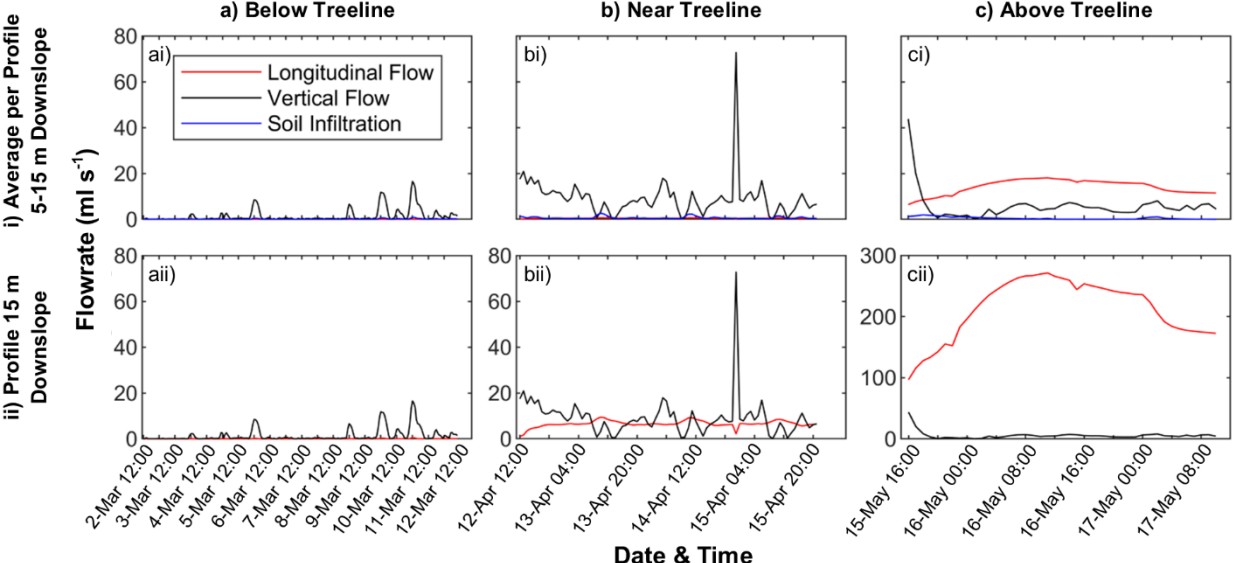

**Figure 5: Results of SnowTOUGH simulations over the entire intensive observations periods (IOPs) for the study sites: a) below treeline (BT), b) near treeline (NT), and c) above treeline (AT). Results shown are i) the average longitudinal flow, vertical flow, and infiltration across the snow-soil interface per m² area of hillslope from 5 m to 15 m downslope in simulated plots, and ii) the longitudinal flow and vertical flowrates at 15 m downslope for a 1 m² area of hillslope.**

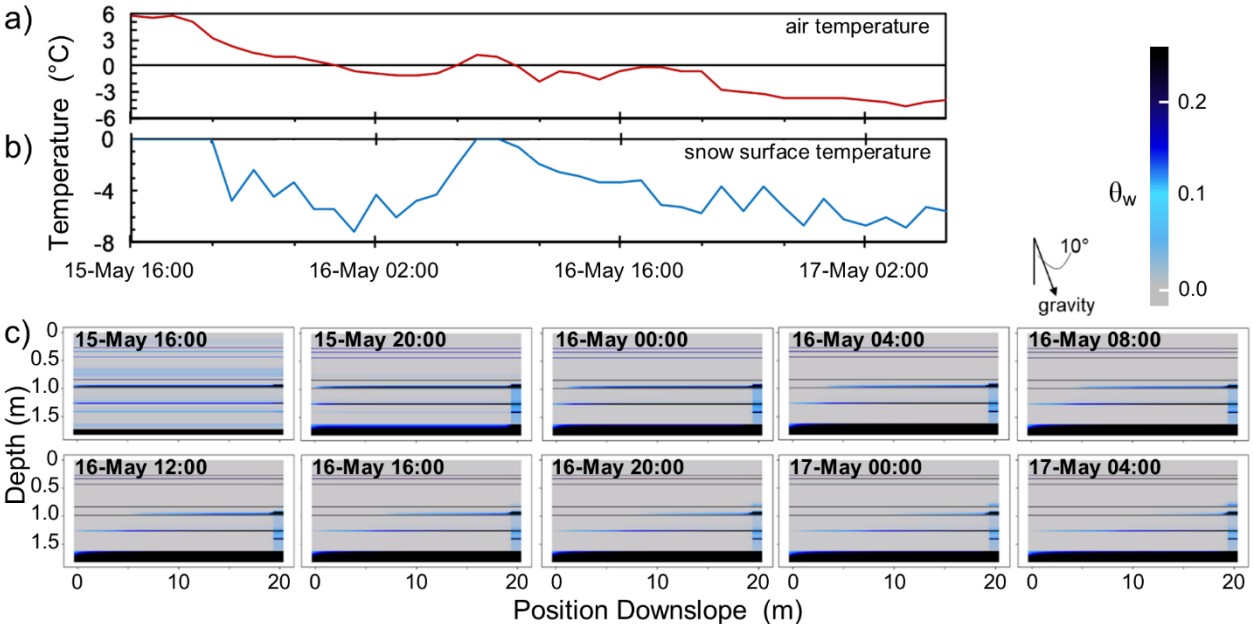

**Figure 6: Time series results of SnowTOUGH simulations for the above treeline (AT) site showing: a) measured air temperature, b) simulated snow surface temperature, and c) volumetric liquid water content (θ_w) distribution at 4-hour intervals.**

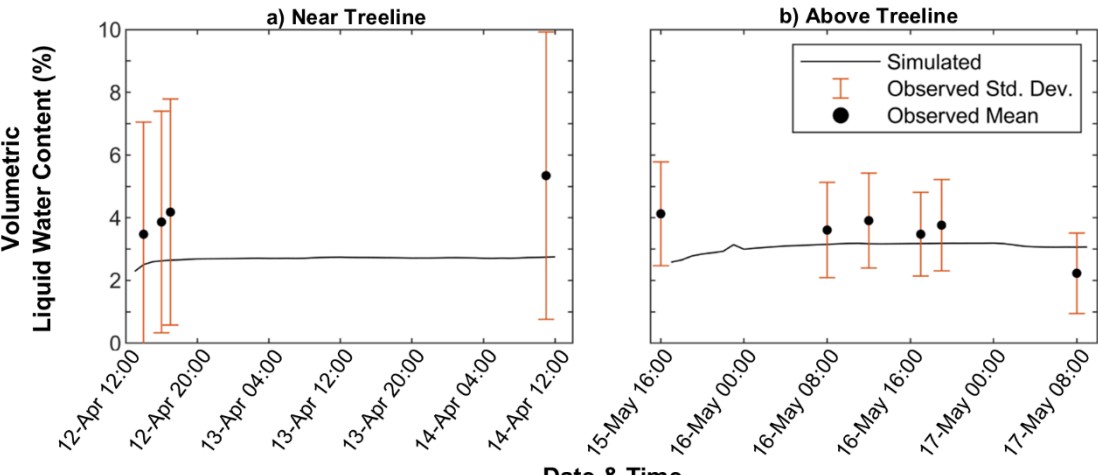

**Figure 7: Comparison of SnowTOUGH simulated volumetric liquid water content ($\theta_w$) and field observations using ground-based remote sensing for the a) near treeline and b)above treeline sites.**