# Peer review of "Two-Dimensional Liquid Water Flow through Snow at the Plot Scale in Continental Snowpacks: Simulations and Field Data Comparisons"

_The Cryosphere, 2020_

## Referee Comment (RC1) · Nander Wever (Referee) · 24 Aug 2020

The authors report on model simulations of lateral water movement in a sloping snowpack. The topic is highly relevant, as it is an open research question to what extend lateral flow in snow modifies the hydrological cycle in snow covered catchments. The coupling of a dedicated snow model with a dedicated water flow model is a unique and very interesting approach. The authors find a much larger lateral flow than vertical flow, suggesting that lateral flow is highly important, even in relatively flat terrain (10 degrees slope angle). Even though the concept of the study is very robust, the manuscript itself falls short at several aspects, and many issues are not, or only weakly discussed. It is

strongly encouraged that the authors thoroughly improve the manuscript. The writing style is of very high quality, with clear figures. However, I think the authors should avoid using the jet-type color scale used in some figures.

**Main concerns:**

The main drawback of the way the study is presented is the lack of validation with the available field data (as for example published by the authors as Webb et al, 2020), except for the observed dye tracer layers showing the lateral flow layers. There must be more validation data available to validate the results of, for example, the SNOW-PACK simulations. This should be worked into the results and discussion sections. For example, does the field data confirm the degree of wetting of the SNOWPACK at the initiation of the field campaigns?

Also, it is reported that: "Initial conditions were provided through manual snow pit observations" (L92), while at the same time, it is reported that: "Within this domain, the iTOUGH2 numerical model simulated the flow of liquid water with time-varying snow layer properties provided by the SNOWPACK model." (L59). That seems contradictory. I have some difficulty reviewing the remainder of the manuscript, since this issue is so unclear. When only material properties are taken from SNOWPACK, without any consideration for the wetting of the snowpack, or the exact layer transitions that are simulated by SNOWPACK (which are a reflection of the driving data to run the simulations), while at the same time observations are used to initiate the iTOUGH2 simulations, then I think that the coupling introduces large errors, simply resulting from the mismatch between simulated and observed stratigraphy. I'm highly confused at this point what exactly has been coupled. In fact, material properties don't vary that much for the couple of days simulated with iTOUGH2, such that when iTOUGH2 can be initialized with observations, the SNOWPACK simulations may not add any information regarding the change in microstructural properties. Maybe it would be a sufficient approach then to have SNOWPACK inform iTOUGH2 about melt rates only, to prevent the problems with potential mismatches between modeled and observed stratigraphy?

All this remains very unclear now.

Furthermore, I struggle to relate earlier reported dye tracer levels to the ones shown here. For example, I assume that the "above treeline" site in Webb et al. (2020) corresponds to the alpine site in this study (it may be a good idea to use the same naming for the different sites as the previous study). Fig. 9(aii) in Webb et al. (2020) shows a dyed layer at the snow/soil interface and around 1.50m inside the snowpack (about 35 cm below the snow surface), both which I can find back in Fig. 3c. However, Fig. 9(aii) in Webb et al. (2020) also shows a layer around 1m above the soil, which seems to be indicated much lower in the snowpack in Fig. 3c. Also, for the treeline site the layers don't seem to correspond. Fig. 9(bii) shows a first dyed layer around 60cm below the snow surface, and another one 60cm above the soil/snow interface, with a third one in between. Fig. 3b shows the highest dyed layer much higher in the snowpack (about 30 cm below the surface) and the lowest about 20cm above the snow/soil interface.

In Webb et al. (2020), it is reported that the SNOWPACK simulations were run using the Richards equation water transport scheme. However, the liquid water content distributions shown in Fig. 2 in Webb et al. (2020) and Fig. 2a in this manuscript, are remarkably homogeneous, even though I found in multiple studies that using Richards equation in SNOWPACK leads to inhomogeneous water distributions. To me, it looks like results from the bucket water transport scheme. The SNOWPACK model also should indicate the hydraulic barriers, since those impede vertical flow, even though the SNOWPACK model does not consider lateral flow.

Many figure panels aren't discussed in the manuscript.

At several instances, it is important to repeat information from earlier published work. For example, the setup and driving data of SNOWPACK simulations should be explained in more detail. Some information is necessary to interpret the results, and only referring to earlier published work is then inadequate in my opinion. For example, it's

СЗ

important to know what the source of precipitation is for SNOWPACK, to understand if the stratigraphy would match local conditions or not. Typically when SNOWPACK is driven with in-situ measured snow depth, it shows better agreement with local stratigraphy, compared to when it is run with rain gauge data. Since there is a large uncertainty in rain gauge data for solid precipitation, individual snow fall events can be severely over- or underestimated. Based on citations, I assume SNOWPACK was run with insitu snow depth data, but I think it's important to repeat that kind of crucial information here. Similarly the use of the canopy module, or a better description of the field sites are important aspects to repeat.

Minor comments:

- I would recommend to use the same terminology as in Webb et al. (2020) to denote the field sites (i.e., above treeline, near treeline, below treeline).

- A discussion of boundary conditions upslope and downslope in iTOUGH2 needs to be added. It seems to be a zero flux boundary to the left and right, such that water accumulates at the downslope boundary. This may also explain the role of the drain mentioned in L91, to prevent water accumulating in the model because of zero flux boundary conditions.

- L77: "was calculated by subtracting theta\_w from rho\_s". Please reformulate, because this doesn't seem to be a calculation that makes sense if theta\_w is not multiplied by the density of water. Furthermore, why not simply multiply theta\_ice with the ice density of 917 kg/m3 to get dry density?

- Eq. 1 and 2: These equations come from Yamaguchi et al. (2012), not Yamaguchi et al. (2010).

- L90: "10-30 cm deep soil". Please denote what kind of soil was prescribed and how this relates to the soil at the study plots. Related: in L138: the lateral flow at the snow-soil interface may be caused by the prescribed soil type. Thus, it is imperative to

discuss this.

- L91: Please explain why deeper soil under deeper snow.

- L91: Please explain the role of the drain in the simulations and in the analysis.

- L102: Please specify if the north or south facing subalpine site was taken.

- L112-116: It's not clear where/how this data is used in the manuscript.

- In L139, it is not clear how the vertical water transport is calculated. It is obviously not homogeneous in the vertical direction, so does this concern the average vertical flow over all layers in the snowpack? I actually think that it is also interesting to relate this number to meltwater input, or water arriving at the snow/soil interface, or water infiltrating into the soil, to put it in broader perspective with respect to hydrological processes.

- L143: "liquid water continued to flow" Please specify if this is true for both SNOWPACK and iTHOUGH2, or that it only concerns the lateral flow in iTHOUGH2.

- L190: liquid water in the snow matrix is not necessarily evidence of flowing water, since the capillary forces will retain some liquid water (the irreducible liquid water content, or residual water content).

- Fig. 3: right panel is labeled "b) Alpine" which should be "c) Alpine"

Feel free to contact me in case anything is unclear in this review.

Nander Wever (nander.wever -at- colorado.edu).

---

## Author Comment (AC1) · 27 Aug 2020

We would like to first express our appreciation to Dr. Nander Wever for reviewing our manuscript. We especially appreciate the constructive criticism of our work and the suggestions for improving the manuscript. We believe that some excellent points are made to improve the final product. Below are our initial responses to the comments with the original comment quoted followed by our response in blue. We are happy to further discuss anything that is not clear.

Comment: "Even though the concept of the study is very robust, the manuscript itself falls short at several aspects, and many issues are not, or only weakly discussed."

Response: We appreciate this broad feedback on our manuscript. I think that we revised an earlier version of the manuscript to focus on brevity and ended up over-editing. Many of the points that you identify are true and more information would likely make the study clearer and offer further discussion and insights into processes. We address these concerns in the other, more specific comments below.

Comment: "The main drawback of the way the study is presented is the lack of validation with the available field data (as for example published by the authors as Webb et al, 2020)"

Response: Looking back over the manuscript, I definitely see your point here. Per my last response, I think we ended up over-editing for brevity and relying a bit too much on the previously published field observations. In the future revision we will certainly include more of the validation. As suggested in your comment, SNOWPACK comparisons do compare well to observed wetting of the snow profile from a bulk perspective (total LWC in entire profile) and we will also check wetness of layers.

Comment: "Also, it is reported that: "Initial conditions were provided through manual snow pit observations" (L92), while at the same time, it is reported that: "Within this domain, the iTOUGH2 numerical model simulated the flow of liquid water with time-varying snow layer properties provided by the SNOWPACK model." (L59). That seems contradictory."

Response: Many of your comments and thoughts on this are true. We apologize for not being clearer in the writing. We can certainly expand on this methods section in revisions. But to clarify in this interactive discussion, we initiated a SNOWPACK simulation with snow pit observations. The hourly output from SNOWPACK are used to define the material properties and introduction/subtraction of any liquid water through melting/freezing in iTOUGH2. Any liquid water that remains and is not re-frozen is transported following Richards'' equation in the iTOUGH2 simulation.

The microstructure of the snow does not change much at the time scale of our study, with the exception of the melting/accumulating layers near the snow surface. iTOUGH2 has not previously been capable of temporally varying properties or the removal of layers, so these upper layers in the snow allowed this new capability to be tested along with pretty minor changes deeper in the snowpack.

Comment: "Furthermore, I struggle to relate earlier reported dye tracer levels to the ones shown here"

Response: These are excellent points and I will make sure to double check that our presented results match with observations previously published. However, it is important to note that there were multiple dye tracer experiment observations at some of the sites reported in previous publications. This will be made more clear in revisions through using the same name of sites and specifying which set of observations are used. We will also double-check that observations are correctly input into figures.

Comment: "In Webb et al. (2020), it is reported that the SNOWPACK simulations were run using the Richards' equation water transport scheme. However, the liquid water content distributions shown in Fig. 2 in Webb et al. (2020) and Fig. 2a in this manuscript, are remarkably homogeneous, even though I found in multiple studies that using Richards' equation in SNOWPACK leads to inhomogeneous water distributions. To me, it looks like results from the bucket water transport scheme"

Response: Thank you for pointing this out. Upon further review of our input files, we did use the bucket scheme for seasonal simulations to have an idea of the temporal evolution of liquid water (i.e. timing of melt, etc.). We then use Richards'' scheme for the shorter time domain of the study periods. We will add more detailed descriptions to explain this and figures that show the results of the Richards'' scheme as implemented in the study. We apologize for this mistake and thank you for noticing this.

Comment: "At several instances, it is important to repeat information from earlier published work. For example, the setup and driving data of SNOWPACK simulations should be explained in more detail."

Response: I agree. I think this is one example where we over-edited for brevity. We will repeat information from previous citations to make the current manuscript stand-alone.

Minor Comments: multiple

Response: Thank you for these comments. The requests for more information pertaining to specific calculations and methods will be included in revisions and directly addressed in the final response to reviews.

If you would like further clarification on specific methods or have further thoughts on how we may improve the presentation of results, we welcome your feedback.

Thank you again for the thorough review.

---

## Referee Comment (RC2) · Anonymous Referee #2 · 14 Sep 2020

In this paper, simulation of liquid water infiltration in the slope snow was discussed. Water infiltration in the slope snow is an important theme for understanding the water infiltration process. The features of this simulation are two: combining SNOWPACK and iTOUGH2, and application of plot scale. Using different resolution scales for parallel and perpendicular direction for slope, it can support both detailed changes of snow layer for vertical direction and plot scale for horizontal direction. This combination of models is good work. However, this paper lacked the validation of longitudinal flow by comparing field data despite the authors working for field observation about longitudinal flow in previous study. As far as reading this paper, only the number of longitudinal flow paths was validated for three field sites. The comparison of existence

of longitudinal flow path is of validation of capillary or hydraulic barrier formation, but not longitudinal flow itself. Since simulated slope flow characteristics (e.g. distance of movement by longitudinal flow) are not authorized by real data, one of the results of this paper, the ratio 250:1 is questionable. Before accepting this paper, validation by quantitative comparison between simulation and field data in terms of longitudinal flow is necessary. Even if the simulation results don't match with the measured result well enough, discussion of the causes of the discrepancies and improvement that will be needed in the future will be useful information for the paper of slope flow with this scale.

Minor comments

P3 L75-82 Please clear what scheme did the author use for the water transport in the SNOWPACK (bucket, Richards equation or dual domain approach). According to Fig. 2a, I guess the bulk scheme was used. In this study, the water infiltration may be estimated by iTOUGH2 part and received little effect by water infiltration scheme of SNOWPACK. But even if so, the type of scheme should be written.

P4 L91-92 The resolution of elements differs 50 times between parallel and perpendicular to the slope. Does it lead any problem for correct simulation due to this large resolution difference?

P4 106-116 This paragraph describes that authors performed the snow pit observation of tracer experiment, measuring water content distribution. These data should be used for quantitative validation especially the distance of the water movement for longitudinal direction. Comparison with these observations enhances the value of simulation result.

P5 L131-135 Snow profile data should be shown in Fig. 3 especially grain size and snow density. These parameters relate the formation of capillary or hydraulic barriers which lead to longitudinal flow.

P5 148-155 The number of longitudinal flows were used for the validation of this model. Although, accuracy of longitudinal flow path means that capillary or hydraulic barrier

was reproduced correctly, it did not mean the slope flow characteristics was repro-duced well. Also, authors should show figures of result of field experiments, not only reference.

P6 173-175, P7 L193-195 In my opinion, neglecting heterogeneity affects the ratio of water flow direction (parallel or perpendicular to slope) rather than the number of lon-gitudinal flows. Heterogeneity sometimes leads the movement to a difficult direction, which leads to decreased the ratio of water flow direction. So the ratio (250:1) has the possibility to be overestimated of the ratio due to neglecting heterogeneity. Further-more, it was not endorsed by field observation.

P7 L197-199, 205-206 Authors have several field data. But they were only used for mention for consistency of the trend. Can the author make quantitative comparison between field data and simulation?

---

## Author Comment (AC2) · 15 Oct 2020

Thank you for the review of our manuscript and I apologize for not responding to this review sooner.

The major comment is that quantitative comparison of the model results to field data are not done. While we agree that this would greatly improve the manuscript, there are some difficulties that arise. For example, our model has a 20 m long domain with no flow upper boundary conditions whereas field data have no such limitations. However, we can likely make some comparisons based on the previously published observations and are happy to spend more time making these comparisons.

[Figure]

Minor comments:

P3 L75-82 Please clear what scheme did the author use for the water transport in the SNOWPACK (bucket, Richards equation or dual domain approach).

Response: We will clarify the scheme in revisions.

P4 L91-92 The resolution of elements differs 50 times between parallel and perpendicular to the slope. Does it lead any problem for correct simulation due to this large resolution difference?

Response: The variability in properties occurs in the vertical direction which is what justifies the difference in resolution. This should not have an impact on the simulation results at the scale of interest.

P4 106-116 This paragraph describes that authors performed the snow pit observation of tracer experiment, measuring water content distribution. These data should be used for quantitative validation especially the distance of the water movement for longitudinal direction. Comparison with these observations enhances the value of simulation result.

Response: We can certainly add some comparisons to these data as suggested.

P5 L131-135 Snow profile data should be shown in Fig. 3 especially grain size and snow density. These parameters relate the formation of capillary or hydraulic barriers which lead to longitudinal flow.

Response: We will add these data in revisions.

P5 148-155 The number of longitudinal flows were used for the validation of this model. Although, accuracy of longitudinal flow path means that capillary or hydraulic barrier was reproduced correctly, it did not mean the slope flow characteristics was reproduced well. Also, authors should show figures of result of field experiments, not only reference.

Response: We will add more comparisons with field data as previously mentioned.

P6 173-175, P7 L193-195 In my opinion, neglecting heterogeneity affects the ratio of water flow direction (parallel or perpendicular to slope) rather than the number of longitudinal flows. Heterogeneity sometimes leads the movement to a difficult direction, which leads to decreased the ratio of water flow direction. So the ratio (250:1) has the possibility to be overestimated of the ratio due to neglecting heterogeneity. Furthermore, it was not endorsed by field observation.

Response: This is one of the reasons that we discuss heterogeneity being a necessary question for future research (Line 181). As for the ratio of 250:1, this occurred many hours after surface melt stopped and many layers naturally drained. field data show liquid water remained in layers of hydraulic barriers, thus this ratio occurs when most vertical flow has stopped and almost only longitudinal flow along the layers remains. This will be clarified in revisions.

P7 L197-199, 205-206 Authors have several field data. But they were only used for mention for consistency of the trend. Can the author make quantitative comparison between field data and simulation?

Response: More comparisons, including quantitative where possible, will occur in revisions.

---

## Author Response (AR1)

Dear Editors and Reviewers,

We would like to thank you for the thorough reviews of our manuscript. We appreciate you all taking the time required to provide the feedback. We have addressed all of the comments in the latest revision of the manuscript and provide responses to each of the comments in the text below. We list the comments in the previous review followed by our response in blue text. Again, thank you.

Sincerely,

Ryan Webb

Editor Decision: Reconsider after major revisions (further review by editor and referees) (26 Oct 2020) by Guillaume Chambon

Comments to the Author:

Dear authors,

I thank you for your constructive answers to referees' comments in the interactive discussion. In view of the intended revisions described in your responses, you are invited to submit a suitably revised manuscript, along with a tracked-change version and point-by-point rebuttal letters. Since both referees asked for major revisions, the new version will be sent to them for a further assessment. Final decision regarding publication will be made based on whether their concerns are effectively addressed. In particular, the issue of providing stronger and more quantitative validations against field data, which is raised by the two referees, shall need to be answered carefully.

Best regards,

Guillaume Chambon / TC Topical Editor

Response: We have provided a response to each comment below. In particular, we have added analyses and figures for improved quantitative validations and insights towards our results.

Anonymous Referee #2

In this paper, simulation of liquid water infiltration in the slope snow was discussed. Water infiltration in the slope snow is an important theme for understanding the water infiltration process. The features of this simulation are two: combining SNOWPACK and iTOUGH2, and application of plot scale. Using different resolution scales for parallel and perpendicular direction for slope, it can support both detailed changes of snow layer for vertical direction and plot scale for horizontal direction. This combination of models is good work. However, this paper lacked the validation of longitudinal flow by comparing field data despite the authors working for field observation about longitudinal flow in previous study. As far as reading this paper, only the number of longitudinal flow paths was validated for three field sites. The

comparison of existence of longitudinal flow path is of validation of capillary or hydraulic barrier formation, but not longitudinal flow itself. Since simulated slope flow characteristics (e.g. distance of movement by longitudinal flow) are not authorized by real data, one of the results of this paper, the ratio 250:1 is questionable. Before accepting this paper, validation by quantitative comparison between simulation and field data in terms of longitudinal flow is necessary. Even if the simulation results don't match with the measured result well enough, discussion of the causes of the discrepancies and improvement that will be needed in the future will be useful information for the paper of slope flow with this scale.

Response: Thank you for your comments. We agree that further validation is necessary, though difficult with the field data that were collected. However, we have added further analysis and results to accomplish this. Additionally, we have re-organized the results that include the 250:1 ratio because this ratio is driven by times of low melt and thus low vertical percolation. We now compare averages over the entire simulated period.

New text concerning these analyses are as follows:

Comparison to field data

Lines 193-200:

[revised manuscript text omitted]

Minor comments

P3 L75-82 Please clear what scheme did the author use for the water transport in the SNOWPACK (bucket, Richards equation or dual domain approach). According to Fig. 2a, I guess the bulk scheme was used. In this study, the water infiltration may be estimated by iTOUGH2 part and received little effect by water infiltration scheme of SNOWPACK. But even if so, the type of scheme should be written.

We have now clarified, added justification for choices, and added another figure.

Lines 83-89:

"Liquid water transport was simulated using the default bucket scheme for full water year simulations and the Richards equation option (Wever et al., 2014) for simulating the intensive observation period (IOP) at each study plot. Full water year simulations were used to offer context to the timing of each IOP relative to peak snow water equivalent (SWE) and snowmelt processes (Fig. 2). For the IOP simulations, initial conditions were provided through manual snow pit observations (Webb et al., 2020; Webb et al., 2018c) so that we could focus our analysis on the intra-snowpack flow of liquid water and comparisons to field observations rather than the accuracy of the SNOWPACK simulated stratigraphy and the potential implications on our results."

P4 L91-92 The resolution of elements differs 50 times between parallel and perpendicular to the slope. Does it lead any problem for correct simulation due to this large resolution difference?

Response: The heterogeneity occurs in the vertical direction and we do not believe that element dimensions have an effect on the results at the scale of our simulations. We did run some initial tests at multiple resolutions with synthetic data and found no effect, but did end up using 25 cm long elements for the purpose of visualizations of results. In the original manuscript, 50 cm long elements were mentioned, we have corrected this as we actually used 25 cm long elements.

If lateral heterogeneity were to be included in future studies then this would have a much larger impact.

P4 106-116 This paragraph describes that authors performed the snow pit observation of tracer experiment, measuring water content distribution. These data should be used for quantitative validation especially the distance of the water movement for longitudinal direction. Comparison with these observations enhances the value of simulation result.

Response: We have added this comparison. Note that we only measured the spatial distribution of bulk water content in the snowpack, not the liquid water profile in each snow pit. The snow pits at the end of the experiments were used to find where the dye tracer had traveled. There were many snow pits dug to do this and so there was not enough time to get full detailed profile information at each location.

P5 L131-135 Snow profile data should be shown in Fig. 3 especially grain size and snow density. These parameters relate the formation of capillary or hydraulic barriers which lead to longitudinal flow.

Response: A new figure has been added. New figure 3 (shown below) shows these data and the following SNOWPACK simulations of the profile.

[Figure]

P5 148-155 The number of longitudinal flows were used for the validation of this model. Although, accuracy of longitudinal flow path means that capillary or hydraulic barrier was reproduced correctly, it

did not mean the slope flow characteristics was reproduced well. Also, authors should show figures of result of field experiments, not only reference.

The results of the field experiments are now shown in two figures. One showing the location of the dye tracers (Fig. 4) and one showing the liquid water content comparisons (Fig. 7). We have also added text to compare field data to simulations for longitudinal flow as previously quoted.

[Figure]

P6 173-175, P7 L193-195 In my opinion, neglecting heterogeneity affects the ratio of water flow direction (parallel or perpendicular to slope) rather than the number of longitudinal flows. Heterogeneity sometimes leads the movement to a difficult direction, which leads to decreased the ratio of water flow direction. So the ratio (250:1) has the possibility to be overestimated of the ratio due to neglecting heterogeneity. Furthermore, it was not endorsed by field observation.

Response: We apologize for not being clear, but we did mean heterogeneity has been previously shown to have minimal influence on the distance of lateral flow from capillary barriers. But you are correct that for permeability barriers it will likely decrease the flow. We have now added text to clarify that these simulations likely overestimate longitudinal flow. We additionally discuss what we believe the correct flows likely are and make recommendations for future studies.

Lines 241-256:

"However, previous studies of capillary barriers at the interface between soil layers have shown that homogenous layer assumptions, as those made in the present SnowTOUGH simulations, capture the average of randomized heterogeneous simulations (Ho and Webb, 1998). The validity of this assumption

for snow should be further studied.  In general, it is likely that the natural heterogeneity of both permeability and capillary barriers will decrease the amount of longitudinal flow simulated in this study. Thus, SnowTOUGH simulations are likely overestimating the amount of longitudinal flow for specific flow paths.

Relative to estimates of EUCA, the AT site simulations overestimated longitudinal flow. However, it is important to note that the field methods used to estimate the EUCA likely underestimate the value because it assumes all diverted liquid water remains in the snowpack at the point of calculation. Additionally, the low melt rates as a result of the incoming storm add uncertainty to the appropriateness of these calculations using snowmelt rates. Therefore, it is likely that the true value of EUCA is between 17 m2 and 51 m2 for the AT site. Conversely, the NT site simulations resulted in a similar amount of longitudinal flow within the plot-scale simulations as field observed EUCA suggests. Considering the underestimation of the number of flow paths simulated at this site and the underestimation of EUCA from field methods as previously mentioned, the true EUCA is likely larger than the SnowTOUGH-simulated longitudinal flow. Additionally, the longitudinal flux for the single flow path is likely overestimated. We recommend the use of snow lysimeters similar to those implemented in Eiriksson et al. (2013) in future studies to further quantify intra-snowpack longitudinal flow for comparison to the SnowTOUGH model."

P7 L197-199, 205-206 Authors have several field data. But they were only used for mention for consistency of the trend. Can the author make quantitative comparison between field data and simulation?

Response: Yes, as discussed in previous comments and shown in quoted revisions. Thank you for your insights provided in your review. These comments certainly improved the quality of the manuscript.

Nander Wever (Referee)

nander.wever@colorado.edu

The authors report on model simulations of lateral water movement in a sloping snowpack. The topic is highly relevant, as it is an open research question to what extend lateral flow in snow modifies the hydrological cycle in snow covered catchments. The coupling of a dedicated snow model with a dedicated water flow model is a unique and very interesting approach. The authors find a much larger lateral flow than vertical flow, suggesting that lateral flow is highly important, even in relatively flat terrain (10 degrees slope angle). Even though the concept of the study is very robust, the manuscript itself falls short at several aspects, and many issues are not, or only weakly discussed. It is strongly encouraged that the authors thoroughly improve the manuscript. The writing style is of very high quality, with clear figures. However, I think the authors should avoid using the jet-type color scale used in some figures.

Response: We have expanded on our descriptions and analysis as described below, and changed the color scheme for a number of the figures.

Main concerns:

The main drawback of the way the study is presented is the lack of validation with the available field data (as for example published by the authors as Webb et al, 2020), except for the observed dye tracer layers showing the lateral flow layers. There must be more validation data available to validate the results of, for example, the SNOWPACK simulations. This should be worked into the results and discussion sections. For example, does the field data confirm the degree of wetting of the SNOWPACK at the initiation of the field campaigns?

We have added more validation as well as clarification. We discuss the model set up clarifications in the next comment, but now quote the added validation analyses.

Comparison to field data

Lines 193-200:

[revised manuscript text omitted]

Also, it is reported that: "Initial conditions were provided through manual snow pit observations" (L92), while at the same time, it is reported that: "Within this domain, the iTOUGH2 numerical model simulated the flow of liquid water with time-varying snow layer properties provided by the SNOWPACK model." (L59). That seems contradictory. I have some difficulty reviewing the remainder of the manuscript, since this issue is so unclear. When only material properties are taken from SNOWPACK, without any consideration for the wetting of the snowpack, or the exact layer transitions that are simulated by SNOWPACK (which are a reflection of the driving data to run the simulations), while at the same time observations are used to initiate the iTOUGH2 simulations, then I think that the coupling introduces large errors, simply resulting from the mismatch between simulated and observed stratigraphy. I'm highly confused at this point what exactly has been coupled. In fact, material properties don't vary that much for the couple of days simulated with iTOUGH2, such that when iTOUGH2 can be initialized with observations, the SNOWPACK simulations may not add any information regarding the change in microstructural properties. Maybe it would be a sufficient approach then to have SNOWPACK inform iTOUGH2 about melt rates only, to prevent the problems with potential mismatches between modeled and observed stratigraphy?

All this remains very unclear now.

Response: We have clarified all of this in the revisions; in fact, we were essentially doing what you were suggesting. We were also initializing SNOWPACK simulations by pit observations, for our observation period. This was to focus on the longitudinal movement of water rather than how well SNOWPACK captures the stratigraphy.

Lines 79-89:

"Simulations were run at hourly timesteps with quality-controlled meteorological observations. Air temperature, relative humidity, wind speed, incoming shortwave radiation, incoming longwave radiation, and snow depth data were used as forcing data for the SNOWPACK simulations. The SNOWPACK canopy module was activated for the below treeline study plot (described below) using physically representative values of leaf area index (4.0 m2 m-2), canopy height (7.0 m), and direct throughfall fraction (0.2, dimensionless). Liquid water transport was simulated using the default bucket scheme for full water year simulations and the Richards equation option (Wever et al., 2014) for simulating the intensive observation period (IOP) at each study plot. Full water year simulations were used to offer context to the timing of each IOP relative to peak snow water equivalent (SWE) and snowmelt processes (Fig. 2). For the IOP simulations, initial conditions were provided through manual snow pit observations (Webb et al., 2020; Webb et al., 2018c) so that we could focus our analysis on the intra-snowpack flow of liquid water and comparisons to field observations rather than the accuracy of the SNOWPACK simulated stratigraphy and the potential implications on our results."

Furthermore, I struggle to relate earlier reported dye tracer levels to the ones shown here. For example, I assume that the "above treeline" site in Webb et al. (2020) corresponds to the alpine site in this study (it may be a good idea to use the same naming for the different sites as the previous study). Fig. 9(aii) in Webb et al. (2020) shows a dyed layer at the snow/soil interface and around 1.50m inside the snowpack (about 35 cm below the snow surface), both which I can find back in Fig. 3c. However, Fig. 9(aii) in Webb et al. (2020) also shows a layer around 1m above the soil, which seems to be indicated much lower in the snowpack in Fig. 3c. Also, for the treeline site the layers don't seem to correspond. Fig. 9(bii) shows a first dyed layer around 60cm below the snow surface, and another one 60cm above the soil/snow interface, with a third one in between. Fig. 3b shows the highest dyed layer much higher in the snowpack (about 30 cm below the surface) and the lowest about 20cm above the snow/soil interface.

[Figure]

In Webb et al. (2020), it is reported that the SNOWPACK simulations were run using the Richards equation water transport scheme. However, the liquid water content distributions shown in Fig. 2 in Webb et al. (2020) and Fig. 2a in this manuscript, are remarkably homogeneous, even though I found in multiple studies that using Richards equation in SNOWPACK leads to inhomogeneous water

distributions. To me, it looks like results from the bucket water transport scheme. The SNOWPACK model also should indicate the hydraulic barriers, since those impede vertical flow, even though the SNOWPACK model does not consider lateral flow.

Response: You are correct and we apologize for this oversight. We have clarified this in the revised manuscript as quoted above in our previous response to one of your comments.

Many figure panels aren't discussed in the manuscript.

At several instances, it is important to repeat information from earlier published work. For example, the setup and driving data of SNOWPACK simulations should be explained in more detail. Some information is necessary to interpret the results, and only referring to earlier published work is then inadequate in my opinion. For example, it's important to know what the source of precipitation is for SNOWPACK, to understand if the stratigraphy would match local conditions or not. Typically when SNOWPACK is driven with in-situ measured snow depth, it shows better agreement with local stratigraphy, compared to when it is run with rain gauge data. Since there is a large uncertainty in rain gauge data for solid precipitation, individual snow fall events can be severely over- or underestimated. Based on citations, I assume SNOWPACK was run with insitu snow depth data, but I think it's important to repeat that kind of crucial information here. Similarly the use of the canopy module, or a better description of the field sites are important aspects to repeat.

Response: We have now added more detailed descriptions of the modeling work to explain these concerns. The SNOWPACK text is quoted in an above response.

Minor comments: - I would recommend to use the same terminology as in Webb et al. (2020) to denote the field sites (i.e., above treeline, near treeline, below treeline).

Response: Agreed, this has been changed in the manuscript.

- A discussion of boundary conditions upslope and downslope in iTOUGH2 needs to be added. It seems to be a zero flux boundary to the left and right, such that water accumulates at the downslope boundary. This may also explain the role of the drain mentioned in L91, to prevent water accumulating in the model because of zero flux boundary conditions.

Response: We have now added the following language to clarify this.

Line 107-109: "The boundary conditions of the upslope and downslope ends of the domain were simulated as no-flow conditions and a drain was modelled at the downslope end to remove excess liquid water that may build up on the no-flow boundary"

- L77: "was calculated by subtracting theta_w from rho_s". Please reformulate, because this doesn't seem to be a calculation that makes sense if theta_w is not multiplied by the density of water. Furthermore, why not simply multiply theta_ice with the ice density of 917 kg/m^3 to get dry density?

Correct, we did multiply by 1000. We actually conducted both calculations as a simple check on some of our calculations. We have edited the text to the theta_ice calculation for ease of readability.

Line 91-92: "The dry density of each snow layer ($\rho_{ds}$) was calculated by multiplying $\theta_i$ by the density of ice (917 kg m-3)."

- Eq. 1 and 2: These equations come from Yamaguchi et al. (2012), not Yamaguchi et al. (2010).

Yes, they do. We used the more recent parameterization. Thank you for catching this, the reference has been fixed.

- L90: "10-30 cm deep soil". Please denote what kind of soil was prescribed and how this relates to the soil at the study plots. Related: in L138: the lateral flow at the snow-soil interface may be caused by the prescribed soil type. Thus, it is imperative to discuss this.

We have added these data as follows:

Lines 105-114:

"The layered snowpack was modelled above a 10–30 cm deep soil, increasing in depth under deeper snow. Deeper soil was modelled under deeper snow to increase pore storage volume available for any infiltrating water released from the snowpack. The boundary conditions of the upslope and downslope ends of the domain were simulated as no-flow conditions and a drain was modelled at the downslope end to remove excess liquid water that may build up on the no-flow boundary (Fig. 1). Soil types for each site are known as silty loam and retention parameters common to this soil type were used. These parameters were a van Genuchten m value of 0.29, a porosity of 0.67, and van Genuchten α value of 0.02 cm-1. Saturated hydraulic conductivity estimates of the soils were taken as the mean of more than 15 mini-disc infiltrometer observations distributed evenly across a 10 m ⬚ 20 m plot at each site. These saturated hydraulic conductivity values for the below treeline (BT), near treeline (NT), and above treeline (AT) sites were 1.36 ⬚ 10-3 cm s-1, 6.93 ⬚ 10-4 cm s-1, and 8.46 ⬚ 10-4 cm s-1, respectively."

- L91: Please explain why deeper soil under deeper snow.

Text has been added as quoted in the previous comment response.

- L91: Please explain the role of the drain in the simulations and in the analysis.

Text has been added as quoted in the previous comment response.

- L102: Please specify if the north or south facing subalpine site was taken.

Text has been added to indicate the aspect of all sites in the study. The new text is:

L132-133: "ranged in elevation from a north facing BT site at 2700 masl in a lodgepole pine forest, a south facing NT site at 3350 masl in a large forest clearing, and a southeast facing AT site at 3500 masl."

- L112-116: It's not clear where/how this data is used in the manuscript.

Response: Further analysis has now been added that compares these data to simulations.

- In L139, it is not clear how the vertical water transport is calculated. It is obviously not homogeneous in the vertical direction, so does this concern the average vertical flow over all layers in the snowpack? I actually think that it is also interesting to relate this number to meltwater input, or water arriving at the

snow/soil interface, or water infiltrating into the soil, to put it in broader perspective with respect to hydrological processes.

Response: We have added the following text to describe this as well as the infiltration information as suggested.

Lines 122-128:

"Flowrate calculations for each simulation was calculated for a 1 m ⬚ 1 m footprint of hillslope such that longitudinal flow is for a 1 m wide section of hillslope summed over the entire depth of the snowpack and vertical flow is for a 1 m2 area on the ground surface summed over the entire depth of the snowpack. These calculations were conducted at a location 15 m downslope and as an average per snow profile for 10 m upslope from this location. Bulk $\theta_w$ values were also calculated for all snow profiles for this same area between 5 m and 15 m downslope for SnowTOUGH simulations to compare to field observations. SnowTOUGH results were analysed for this 10 m length of hillslope to eliminate boundary effects of the upslope and downslope boundary conditions on analyses."

- L143: "liquid water continued to flow" Please specify if this is true for both SNOWPACK and iTHOUGH2, or that it only concerns the lateral flow in iTHOUGH2.

Response: We have now added text stating that the flow is predominantly in the longitudinal direction to make it clear we are discussing the SnowTOUGH simulations. However, you can also see in the new figure of SNOWPACK simulations that liquid water remains and slowly moves vertically as well.

- L190: liquid water in the snow matrix is not necessarily evidence of flowing water, since the capillary forces will retain some liquid water (the irreducible liquid water content, or residual water content).

Response: Agreed. We have now added text discussing this.

L261-262: ". Furthermore, multiple layers retained liquid water that will be more readily available for transport during later melt events"

- Fig. 3: right panel is labeled "b) Alpine" which should be "c) Alpine"

Thank you for pointing this out, it has been corrected.

---

## Author Response (AR2)

Dear Editors and Reviewers,

We would like to thank you for reviewing the revised version of our manuscript. We appreciate you taking the time required to provide the feedback a second time. We have addressed all of the comments in the latest revision of the manuscript and provide responses to each of the comments in the text below with quoted revised text and line numbers, where appropriate. We list the comments followed by our response in blue text. Again, thank you.

Sincerely,

Ryan Webb

**Editor Decision: Publish subject to minor revisions (review by editor)** (20 Jan 2021)
by Guillaume Chambon
Comments to the Author:
Dear authors,

I have now received the feedback of two referees on your revised manuscript. They are both happy with the revisions and the responses brought to their comments, and recommend publication of the paper. They do however suggest a small number of additional corrections, mainly of technical nature. Please consider addressing these last issues before submitting the final version of your paper.

Best regards,
Guillaume Chambon / TC Topical Editor

Response:
Thank you for your work on this manuscript. We have now addressed all of the comments as detailed below and made appropriate revisions.

Report #1
Nander Wever, nander.wever@colorado.edu

Suggestions for revision or reasons for rejection (will be published if the paper is accepted for final publication)
The authors provided a thoroughly revised manuscript, which can be accepted for publication after taking the comments below into consideration, which are mostly of technical nature.

1) L88/89: "For the IOP simulations, initial conditions were provided through manual snow pit observations (Webb et al., 2020; Webb et al., 2018c)" It's not a trivial task to start SNOWPACK simulations from snow pit observations, since you need parameters such as bond size, sphericity and dendricity, which is typically not available from manual snow pit observations. Please provide more details here.

Response:
We tried to be as consistent as possible with the way SNOWPACK represents these parameters with the measurements we had taken (provided in text as new lines 89–93).
- We assigned liquid water fraction values based on field observations (very wet/slush = 0.07, wet = 0.05, little water = 0.03, sticky = 0.01) that were also confirmed by GPR bulk estimates.

- Because the snow was old at time of observation, we set dendricity to 0 and sphericity to 1
- Bond radius was set to be 0.3*rg where rg is the measured grain radius
  - If multiple grain measurements were given in a layer, rg was the average of all values
- Grain types were coded based on Fig. 8 in Lehning et al. (2002)

We've since added all the model data to a Mendeley Data repository with the link added in the acknowledgements: http://dx.doi.org/10.17632/9wjjvsy82p.1 (will update when paper accepted, repo can be previewed here: https://data.mendeley.com/datasets/9wjjvsy82p/draft?a=8596c963-250a-4f1f-b24f-7e6c88b8d8d8)

New text added to manuscript:
Lines 89-94: "We revised input files for these simulations to be as consistent as possible with SNOWPACK's representation of measured and non-measured parameters. For example, we set liquid water fraction values based on snow pit observations (very wet/slush = 0.07, wet = 0.05, little water = 0.03, sticky = 0.01; Bradford et al., 2009; Techel & Pielmeier, 2011; Webb et al.,2018c). We also defined sphericity to be 1 and dendricity to 0 as most of the snow had metamorphosed by the time of observation, and we set grain types based on Lehning et al. (2002). We estimated bond radius as 30% of the measured grain radius."

2) L113: this porosity is very high for typical soil. Is it maybe (1-porosity), i.e., soil volumetric content, that is reported here?

Response:
The porosity is on the higher end. However, this will allow for more soil storage to avoid any soil saturation and exfiltration from impacting our results. We maintained similar hydraulic properties elsewhere based on field collected hydraulic conductivity data.

New text:
Line 117: "The slightly high porosity was chosen to allow for additional soil moisture storage, if necessary"

3) L118: To prevent confusion, maybe write: "Similar to the SNOWPACK simulations used for the IOP, ", since there are also the seasonal simulations using SNOWPACK that are not initialized using field observations.

Response:
Agreed and revised.

4) L125: "Flowrate calculations for each simulation was calculated for a 1 m x 1 m footprint of hillslope." Awkward sentence, please reformulate.

Response:
Revised. New text:
Lines 128-130: "Flowrates for each simulation were calculated for a 1 m x 1 m footprint of hillslope. Thus, longitudinal flow is for a 1 m wide section of hillslope summed over the entire depth of the snowpack and vertical flow is for a 1 m$^2$ area on the ground surface summed over the entire depth of the snowpack."

5) Section 4.1 should offer a more detailed explanation of what can be seen in the figures 2 and 3 (see my comment in my earlier review that figure panels are poorly discussed, which I think is still the case). Examples are to mention dates explicitly when the first wetting can be found in for example the BT site, the simulated capillary/hydraulic barriers for the NT and AT site that impact water flow, etc. Section 4.1 is really short, so there is room to really set the stage and make sure that readers understand how these sites differ in simulated snow cover.

Response:
We agree that more could be added, though are trying to keep the text brief. The timing of melt is currently included in section 4.1 in lines 156-159. New text added is as follows:

Lines 165-172: "All three site simulations expressed diurnal melt cycles, including the retention of liquid water overnight. The BT simulations resulted in diurnal melt cycles that ranged from 0-1.1 mm hr-1. The NT simulations resulted in melt rates ranging from 0-3.9 mm hr-1, and the AT simulations resulted in melt rates ranging from 0-2.7 mm hr-1. Additionally, the NT and AT SNOWPACK simulations display the occurrence of hydraulic barriers that persist throughout the entire melt season (Fig. 3). These results also highlight the increased stratigraphy and formation of ice lenses that occur at higher elevations and impact liquid water flow processes (e.g. Webb et al., 2018b). Thus, the IOP SNOWPACK simulations resulted three different snowpack conditions that varied in melt-freeze cycles, metamorphism, and snow accumulation/disappearance throughout the profiles."

6) L164: grain size or density? I see mostly density changing during melt, not so much grain size. Or do the authors mean that grain size at the surface changes because deeper layers (the one with an ice layer) come at the surface? Or is maybe the color bar not really clear here?

Response:
Grain size is correct. When initialized with snow pit data, SNOWPACK-simulated grain sizes were highly persistent (I.e., little observable change). We mean that grain sizes changed as the grains melted away and disappeared.

7) L165: "All three sites resulted in simulated diurnal melt cycles" Awkward sentence, please reformulate.

Response:
Changed to "All three site simulations expressed diurnal melt cycles"

8) L174-175: Something doesn't fit here. After discussing the BT site (which is the lowest of all), the text continues by talking about "These simulations display...", followed by a sentence discussing the high elevation sites. Please rephrase.

Response:
Agreed. Edited for clarity. Now, after discussing the BT site we added text to transition the discussion to the higher elevation sites. The new text reads:

Lines 179-180: "This dominance of vertical flow contrasts with the simulations of the two higher elevation snowpacks, which display the higher occurrence of hydraulic barriers as a result of the more complex stratigraphy"

9) L174: Figure 5 should be introduced when mentioning the 1.5 ml/s statistic.

Response:
Added

10) Fig. 5: Why is Fig. 5ai,bi and ci not depicting the middle of the profile, or at 5m from the top, to make a more meaningful comparison with the plots shown in 5aii,bii and cii?

Response:
The top row is the average for the middle of the hillslope (from 5 m to 15 m downslope) as mentioned in the caption. This shows an average proportion of lateral diversion per profile that accumulates at the 15 m downslope location. We have modified the label to indicate this and hopefully reduce any confusion.

11) L189: Please mention the specific date for the onset of the storm here to make it easier to compare to the figures.

Response:
Revised to specify "the afternoon of May 15"

12) L205: "Comparisons of SnowTOUGH to field observations varied." I assume you mean that the comparisons showed varying results, not that the comparisons varied?

Response:
Yes, revised to read: "Comparisons of SnowTOUGH to field observations indicated varying results based on site and parameter of interest"

13) Section 4.1/4.2: simulated melt rates are mentioned in Section 4.3, but I think the interpretation of Section 4.2 is helped when melt rates are reported there, to put them in context with the simulated flow rates.

Response:
We have now added melt rates from the SNOWPACK simulations to section 4.1 as quoted in a previous response.

Report #2
Hiroyuki Hirashima, hirasima@bosai.go.jp

Suggestions for revision or reasons for rejection (will be published if the paper is accepted for final publication)

This paper has been improved by corrections that I requested in the first review. In addition, a comparison with actual measurements of longitudinal flow has been included, making the paper more informative than the first manuscript. Since the consistency of the model was confirmed by comparison with observations, I think it is suitable to publish the paper in TC.

For the comparison with actual observations which is newly added in 4.3, I have comments on it. This is not a requirement for acceptance, so please refer to it when preparing your final draft.

Line 193-200

In the comparison, the measured water content was larger than the simulation which means underestimation of simulation. Considerable causes of this are underestimation of the retained moisture content, percentage of wet snow, or size of water saturated layer due to capillary or hydraulic barrier in the simulation. Do you have any opinion on which one affected this underestimation? If the water saturated layer was underestimated in the simulation, it may lead smaller longitudinal flow. Since this is a comparison with the field observation, it may be difficult for detailed analysis. But If you can describe possible views, it will help to raise future research themes.

Response:
I think this is part of the text that mentions the high mean values are partly driven by the locations with liquid water contents as high as 20%. Also, this is within 2% of observations which is within the estimated error of the techniques, so it could be a possible error in the observations or simulations. I think it is likely a combination of uncertainty in both the modeling and the field techniques. That is why I try to end the paragraph with a statement about the error of the field methods.

Line 210-222

This paragraph was most required in my review. I think the detailed comparison process had better been written because it took a while to understand that EUCA and simulated longitudinal flow per melt rate was compared from this paragraph.
This comparison is desirable to perform in multiple environments and data in the future. In that sense, this paragraph is useful information as a method and first comparison example.

Response:
Thank you for your previous request for this information. We agree that this comparison has improved the manuscript.